# Therapeutic Strategy for Patients with Concomitant Pulmonary Artery Hypertension and Hypertrophic Obstructive Cardiomyopathy: A Rare Case Report

**DOI:** 10.3390/medicina59020401

**Published:** 2023-02-18

**Authors:** Toshihide Izumida, Teruhiko Imamura, Shuhei Tanaka, Shuji Joho, Koichiro Kinugawa

**Affiliations:** Second Department of Medicine, University of Toyama, Toyama 930-0194, Japan

**Keywords:** hemodynamics, collagen disease, heart failure

## Abstract

Combined cases of hypertrophic obstructive cardiomyopathy (HOCM) and pulmonary arterial hypertension (PAH) are rare and have a management dilemma. Although preload is crucial in the management of HOCM, anti-PAH agents dramatically change the preload, leading to improving or worsening heart failure in patients with HOCM. We had a 74-year-old woman with Sjogren-syndrome-associated PAH. Her heart failure worsened following the initiation of anti-PAH agents due to an incremental preload on the left ventricle, whereas HOCM clinically developed following the termination of anti-PAH agents and progressing anorexia due to the progression of the left ventricular outflow obstruction. Careful monitoring of the left ventricular outflow obstruction during initiation/termination of anti-PAH agents and medical intervention to the HOCM are highly recommended.

## 1. Introduction

Combined cases of hypertrophic obstructive cardiomyopathy (HOCM) and pulmonary arterial hypertension (PAH) are rare and have a management dilemma [1]. An appropriate amount of preload is crucial in the management of HOCM. Initiation/withdrawal of anti-PAH agents affect the amount of preload on the left ventricle and might have negative/positive impacts on HOCM [2,3]. Thus, therapeutic strategy for those with HOCM and PAH remains unestablished.

We had a patient with Sjogren-syndrome-associated PAH. The initiation of anti-PAH agents worsened her heart failure. Clinical symptoms and signs of HOCM developed following the termination of anti-PAH agents. We struggled to diagnose the cause of the heart failure during anti-PAH-agent treatment and how to manage these comorbidities that required opposite treatments.

## 2. Case Presentation

### 2.1. Initial Diagnosis of PAH

We present here a 74-year-old woman. Eleven years ago, the patient was diagnosed with PAH according to the results of the right heart catheterization: elevated mean pulmonary arterial pressure (34 mmHg) and pulmonary vascular resistance (4.1 Wood units), normal pulmonary arterial wedge pressure (15 mmHg), and preserved cardiac output/cardiac index (4.6 L/min and 3.6 L/min/m^2^). Computed tomography of the chest, ventilation-perfusion scintigraphy, and pulmonary arteriography found no abnormalities. She was diagnosed with Sjogren syndrome due to her symptoms of dry eye and dry mouth, the existence of anti Ro/SSa antibody and impaired salivary gland function. However, she had a previous history of steroid-associated necrosis of the talus. Thus, she initiated sildenafil 60 mg/day to treat her PAH, instead of steroids.

### 2.2. Following the Initiation of Anti-PAH Agents

Combination therapy for PAH was performed using sildenafil, macitentan, and riociguat (Figure 1). However, plasma B-type natriuretic peptide (BNP) levels increased gradually together with the worsening of the dyspnea on exertion, categorized as the New York Heart Association function class II. Annual follow-up transthoracic echocardiography found gradually progressive myocardial hypertrophy. Considering suspected hypertensive cardiomyopathy, irbesartan was introduced (Figure 1). Three years ago, auto-antibodies regarding the Sjogren syndrome got negative without any specific interventions.

Two months prior to the index hospitalization, plasma B-type natriuretic peptide levels exceeded 400 pg/mL and we terminated anti-PAH agents to reduce preload and unload the left ventricle. One month after the termination of anti-PAH agents, a transthoracic echocardiography found an accelerated blood flow in the left ventricular outflow tract, with a Vmax of 3.4 m/s with flow separation and moderate mitral regurgitation due to systolic anterior motion of the mitral valve (Figure 2A). We initiated bisoprolol 2.5 mg/day. Nevertheless, her fatigue, dyspnea on exertion, and lightheadedness gradually worsened. Her New York Heart Association function class got categorized as Ⅳ.

### 2.3. On Admission

On admission, her blood pressure was 85/55 mmHg, her heart rate was 50 bpm, her saturation was 99% (room air), and her body weight was 42.4 kg. The mid-systolic murmur was heard at the left sternal border, not radiating to the neck. She had no peripheral edema. The plasma B-type natriuretic peptide level was 1641 pg/mL and serum creatinine level was 2.35 mg/dL.

A chest X-ray showed slight cardiomegaly without the bilateral pleural effusions and congestion. Electrocardiography showed sinus rhythm and a negative T wave in I, II, aVF, and V3–6. Transthoracic echocardiography displayed 50 mm of the left atrial diameter, 15 mm of the interventricular septal, 14 mm of the posterior wall of left ventricle, 40 mm of the left ventricular end-diastolic diameter, and 79% of the left ventricular ejection fraction. The trans-mitral flow pattern had an impaired relaxation, with an E/A ratio of <1.0 and the existence of an L-wave. The inferior vena cava was collapsed and the estimated right atrial pressure was <3 mmHg. Of note is that the left ventricular outflow tract obstruction and moderate mitral regurgitation due to systolic anterior motion of the mitral valve were observed.

### 2.4. In-Hospital Course

Given all together, we suspected the physiology of hypertrophic obstructive cardiomyopathy (HOCM), which was manifested by a decline in preload on the left ventricle by the termination of anti-PAH agents, and continued diuretics despite the progression of anorexia. We terminated diuretics and initiated intravenous fluid supplementation.

Right heart catheterization showed 42/14/26 mmHg of pulmonary artery pressure, 7 mmHg of mean right atrial pressure, 10 mmHg of pulmonary artery wedge pressure, and 2.9 L/min/m^2^ of cardiac index. The left ventricular end-diastolic pressure was 23 mmHg. The left ventricular pressure gradient was 56 mmHg at rest (Figure 3A). The Brockenbrough’s phenomenon was positive (Figure 3B). An endo-myocardial biopsy showed no obvious abnormalities suggestive of secondary cardiomyopathy. We diagnosed her with HOCM.

### 2.5. Following Index Discharge

Following the initiation of cibenzoline succinate 50 mg/day, she was discharged on day 15. At one month following the index discharge, the left ventricular outflow tract obstruction and mitral regurgitation due to systolic anterior motion of the mitral valve disappeared (Figure 2B). Although the corrected QT interval was extended from 436 to 482 milliseconds after cibenzoline administration, further prolongation of the corrected QT interval was not observed during the follow-up period. At two month following the index discharge, her New York Heart Association function class was II, plasma B-type natriuretic peptide was 578 pg/mL, and serum creatinine was 1.06 mg/dL. Her fluid status remained stable, without any peripheral edema without any diuretics.

## 3. Discussion

### 3.1. Management of Sjogren-Syndrome-Associated PAH

Sjogren-syndrome-associated PAH has rarely been reported thus far, whereas several studies reported PAH induced by other collagen diseases [4,5]. Detailed pathogenesis of Sjogren-syndrome-associated PAH remains uncertain given the lack of animal models. Endothelial nitric oxide synthase abnormality and chronic inflammation seem to be associated with the development of this syndrome [6,7].

A therapeutic strategy has not yet been established for this syndrome, but immunosuppressive agents are empirically utilized [6]. Given the history of idiopathic osteonecrosis, our patient did not receive steroid therapy. Nevertheless, auto-antibodies became negative without any specific interventions.

### 3.2. HOCM and PAH

The degree of preload is crucial for the management of HOCM. In general, aggressive dehydration is contraindicated in this cohort. Such an intervention facilitates the outflow obstruction and reduces the cardiac output. On the contrary, excessive volume loading increases intra-cardiac pressure and further impairs the cardiac diastolic function [8]. Anti-PAH agents dilate the pulmonary artery vasculature and decrease the pulmonary artery pressure, while increasing preload on the left ventricle on the other hand [9]. Thus, the management of HOCM in patients with concomitant PAH is challenging [1].

In our patient with potential diastolic dysfunction, the initiation of anti-PAH agents increased the forward flow toward the left ventricle and induced congestive heart failure. On the contrary, termination of multiple anti-PAH agents and progressive anorexia, as well as ongoing diuretics, dramatically reduced preload on the left ventricle and manifested the left ventricular outflow obstruction, resulting in the clinical diagnosis of HOCM.

### 3.3. Recommended Clinical Management

In patients with potential diastolic dysfunction, irrespective of the definite diagnosis of HOCM, it may be recommended to monitor the degree of the left ventricular outflow obstruction during titration/termination of anti-PAH agents. Drastic changes in the preload might manifest occult HOCM. Instead, an appropriate adjustment of preload, using the right heart catheterization and aggressive therapeutic intervention to HOCM, should be prioritized using beta-blockers and cibenzoline, which are effective for reducing the pressure gradient at both rest and during exercise, not through the change of preload and afterload, but through the decreased contraction force, as we did.

For symptomatic drug-resistant HOCM patients, septal reduction therapy should be aggressively considered. Generally, in younger patients, surgical intervention would be preferable. Alcohol septal ablation is alternative in older patients or those with severe PAH. Stabilization of heart failure might improve pulmonary congestion and decrease pulmonary artery pressure [1]. Anti-PAH agents can be added safely if necessary.

## 4. Conclusions

We reported combined case of HOCM and Sjogren-syndrome-associated PAH. The aggressive therapeutic intervention to HOCM should be prioritized in those patients. 

## Figures and Tables

**Figure 1 medicina-59-00401-f001:**
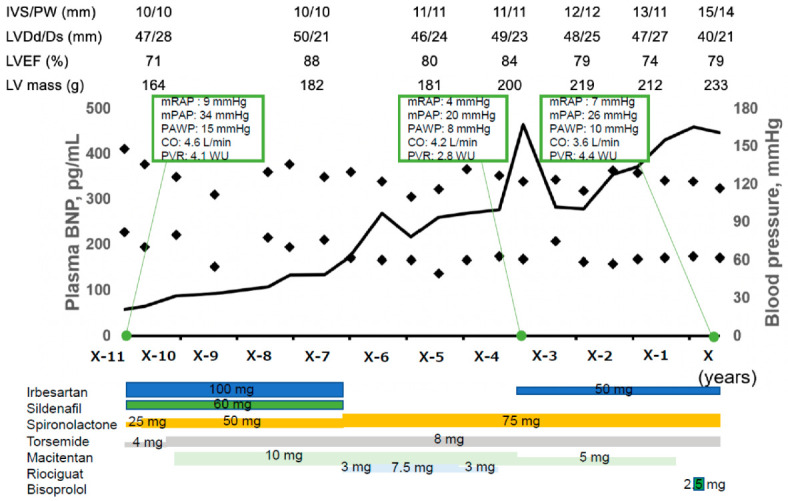
Clinical course before index hospitalization. The patient was diagnosed with PAH according to the results of right heart catheterization. Combination therapy for PAH was performed using sildenafil, macitentan, and riociguat. However, plasma BNP level increased gradually. Annual follow-up transthoracic echocardiography found gradually progressive myocardial hypertrophy despite of initiation of irbesartan and spironolactone and well-controlled blood pressure. IVS, interventricular septal; PW, posterior wall; LVDd, left ventricular diastolic diameter; LVDs, left ventricular systolic diameter; LVEF, left ventricular ejection fraction; BNP, B-type natriuretic peptide; mRAP, mean right atrial pressure; mPAP, mean pulmonary arterial pressure; PAWP, pulmonary arterial wedge pressure; CO, cardiac output; PVR, pulmonary vascular resistance.

**Figure 2 medicina-59-00401-f002:**
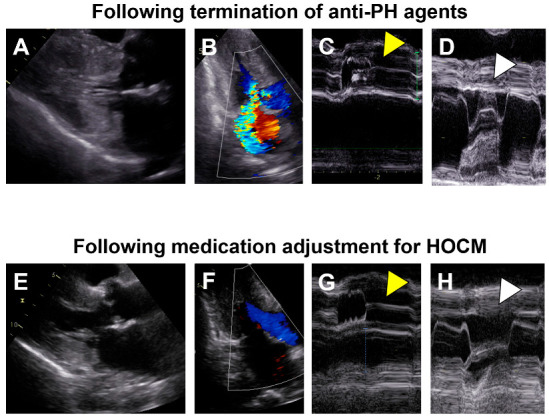
Transthoracic echocardiography images following termination of anti-PH agents (**A**–**D**) and those following medication adjustment for HOCM (**E**–**H**). (**A**) Diffuse left ventricular hypertrophy is obvious; (**B**) An accelerated blood flow in the left ventricular outflow tract with a Vmax of 3.4 m/s and moderate mitral regurgitation probably due to systolic anterior motion of the mitral valve; (**C**) Left ventricular outflow tract obstruction with aortic valve semi-closure (yellow arrow head); (**D**) Systolic anterior motion of the mitral valve (white arrow head); (**E**) Left ventricular hypertrophy got milder; (**F**) Neither obvious left ventricular outflow tract obstruction nor mitral regurgitation; (**G**) No obvious aortic valve semi-closure (yellow arrow head); (**H**) No obvious systolic anterior motion of the mitral valve (white arrow head). (**A**,**B**,**E**,**F**) were obtained at B-mode and others were obtained at M-mode. PH, pulmonary hypertension; HOCM, hypertensive obstructive cardiomyopathy.

**Figure 3 medicina-59-00401-f003:**
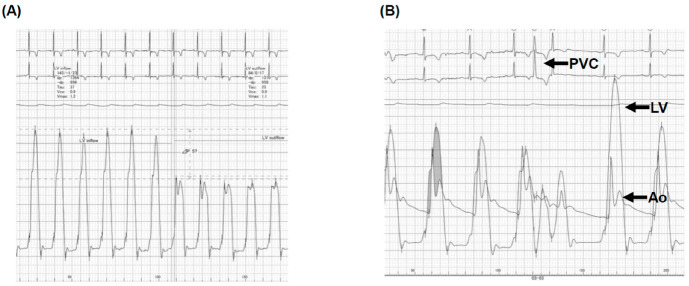
Right heart catheterization at rest (**A**) and at the time of Brockenbrough phenomenon (**B**). (**A**) At rest, the left ventricular pressure gradient was 56 mmHg; (**B**) At the time of the Brockenbrough phenomenon, a significant pressure gradient between LV and Ao, following the PVC, was observed. PVC, premature ventricular contraction; LV, left ventricle; Ao, aorta. (**A**) Right heart catheterization showed 56 mmHg of the resting left ventricular pressure gradient and (**B**) the Brockenbrough phenomenon, which is the significant gradient between the left ventricular (LV) and aortic pressure (Ao) on the beat post-PVC, was positive.

## Data Availability

Data are available from the corresponding author upon reasonable request.

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
