# Peer review of "Therapeutic Strategy for Patients with Concomitant Pulmonary Artery Hypertension and Hypertrophic Obstructive Cardiomyopathy: A Rare Case Report"

_medicina, 2023, doi:10.3390/medicina59020401_

Round 1
Reviewer 1 Report
The authors present a case study of a patient with pulmonary arterial hypertension (PAH) associated with possible Sjogren's syndrome and hypertrophic obstructive cardiomyopathy (HOCM) in a 74-year-old woman. Treatment of PAH increased preload to the LV, unmasking clinical heart failure. When anti-PAH therapy was stopped, she developed symptoms of HOCM with systolic anterior motion of the mitral valve, mitral regurgitation, and evidence of LV outflow obstruction. Treatment with cibenzoline succinate improved the obstruction and mitral regurgitation. Overall, this is an interesting case, but some key data about the case are missing, and several areas should be clarified.
1. LVOT obstruction can be seen in concentric hypertrophy caused by hypertension (hypertensive hypertrophic cardiomyopathy). Do the authors think this was the likely diagnosis as opposed to an inherited myopathy? This is mentioned in the report but not explained well. The ARB and beta blocker apparently did not help her condition, and she had a eventually had hypotension with a blood pressure of 85/55. Were these medications stopped for good at this time? Or was this all due to diuretics? Did she have an AKI associated with this?
2. Please report the right atrial pressure with the other right heart catheterization results in Section 2.4. How do the authors explain the elevated BNP in the setting of low filling pressures? Please report the serum creatinine or GFR. Could CKD have contributed to the high BNP in the setting of lower filling pressure? The diuretics were stopped during the admission. Were they ever restarted later, or did she not need them?
3. Regarding the echocardiogram at the time of hospital admission, please report the septal and lateral e' velocities. Please confirm that the patient was determined to have grade 1 diastolic dysfunction. Please include the septal and lateral LV wall thickness measurements in Section 2.3. Were these increased from the prior echocardiograms? Did these thicknesses decrease after the patient had a clinical improvement?
4. In Section 2.5, was the decreased BNP associated with a decreased creatinine?
5. The disappearance of the antibodies over time is curious. Only the result for SS-A was noted. Was SS-B evaluated, as well? SS-B is more specific for Sjogren's syndrome. Are you sure she really had Sjogren's syndrome? Maybe the salivary gland issue was related to something else. Did she have dry eyes, as well?
6. Cibenzoline is not commonly used in the U.S. in the current era for HCM. Disopyramide is more commonly used. Cibenzoline can prolong the QRS and QT interval. Please provide the effect of the medication on these ECG parameters. Is this medication commonly used by the authors for this condition?
7. English editing is recommended.
Author Response
Reviewer 1
General comment.
The authors present a case study of a patient with pulmonary arterial hypertension (PAH) associated with possible Sjogren's syndrome and hypertrophic obstructive cardiomyopathy (HOCM) in a 74-year-old woman. Treatment of PAH increased preload to the LV, unmasking clinical heart failure. When anti-PAH therapy was stopped, she developed symptoms of HOCM with systolic anterior motion of the mitral valve, mitral regurgitation, and evidence of LV outflow obstruction. Treatment with cibenzoline succinate improved the obstruction and mitral regurgitation. Overall, this is an interesting case, but some key data about the case are missing, and several areas should be clarified.
Response.
We express our great appreciation for the reviewer’s critical suggestions and comments. We attempted our best to answer all the comments and revise our draft. We also corrected typographs and grammatical errors. We believe that the revised draft would be suitable for the publication.
Comment 1.
LVOT obstruction can be seen in concentric hypertrophy caused by hypertension (hypertensive hypertrophic cardiomyopathy). Do the authors think this was the likely diagnosis as opposed to an inherited myopathy? This is mentioned in the report but not explained well. The ARB and beta blocker apparently did not help her condition, and she had eventually had hypotension with a blood pressure of 85/55. Were these medications stopped for good at this time? Or was this all due to diuretics? Did she have an AKI associated with this?
Answer 1.
We appreciate your comment. As you pointed out, we initially suspected hypertensive cardiac hypertrophy, because of age and clinical course. However, our patient developed diffuse myocardial hypertrophy, despite of well-controlled blood pressure and treatment with angiotensin II receptor blockers, mineralocorticoid receptor antagonists, and anti-PAH agents that should have inhibited myocardial hypertrophy. Therefore, we performed endo-myocardial biopsy and evaluated the secondary cardiomyopathy.
ARB was restarted to block the activation of RAS system and beta blocker was continued to treat HOCM. MRA and loop diuretics have been withdrawn to date and the condition is stable. Combination effect of decreased preload and the development of LVOT obstruction due to diuretics resulted in low cardiac output and induced acute kidney injury. We added the below sentence in manuscript.
Before 1.
CASE PRESENTATION
Following index discharge:
None
After 1.
CASE PRESENTATION
Following index discharge:
Her fluid status remained stable without any peripheral edema without any diuretics.
Comment 2.
Please report the right atrial pressure with the other right heart catheterization results in Section 2.4. How do the authors explain the elevated BNP in the setting of low filling pressures? Please report the serum creatinine or GFR. Could CKD have contributed to the high BNP in the setting of lower filling pressure? The diuretics were stopped during the admission. Were they ever restarted later, or did she not need them?
Answer 2.
Thank you for your comment. We considered the reason regarding the elevated BNP to be the elevated LVEDP (23 mmHg) due to LVOT obstruction. Although diuretics has not been restarted since then, considering negative cascade by the preload reduction, her condition was stable and she has not needed diuretics.
We added the data of hemodynamics and the serum creatinine to our manuscript.
Before 2.
CASE PRESENTATION
On admission:
None
In-hospital course:
Right heart catheterization showed 42/14/26 mmHg of pulmonary artery pressure, 10 mmHg of pulmonary artery wedge pressure, and 2.9 L/min/m2 of cardiac index. Left ventricular pressure gradient at rest was 56 mmHg (Figure 3A).
After 2.
CASE PRESENTATION
On admission:
Serum creatinine was 2.35 mg/dL.
In-hospital course:
Right heart catheterization showed 42/14/26 mmHg of pulmonary artery pressure, 7 mmHg of mean right atrial pressure, 10 mmHg of pulmonary artery wedge pressure, and 2.9 L/min/m2 of cardiac index. Left heart catheterization showed that left ventricular end-diastolic pressure was 23 mmHg.
Comment 3.
Regarding the echocardiogram at the time of hospital admission, please report the septal and lateral e' velocities. Please confirm that the patient was determined to have grade 1 diastolic dysfunction. Please include the septal and lateral LV wall thickness measurements in Section 2.3. Were these increased from the prior echocardiograms? Did these thicknesses decrease after the patient had a clinical improvement?
Answer 3.
Thank you for your comment. The septal and lateral e’ velocities were 4.9 and 2.9 m/sec. Although our case would be determined to have grade 2 diastolic dysfunction and the elevated left atrial pressure according to guidelines, hemodynamics showed normal pulmonary artery wedge pressure. It might be the limitation of measurement using echocardiography. As shown in Figure 1, interventricular septal and posterior wall of left ventricle was 15 and 14 mm and increased from the prior echocardiography. These thickness did not decrease, after the patient had a clinical improvement. We revised our manuscript.
Before 3.
CASE PRESENTATION
On admission:
Transthoracic echocardiography displayed 50 mm of left atrial diameter, 40 mm of left ventricular end-diastolic diameter, and 79% of left ventricular ejection fraction. Trans-mitral flow pattern was impaired relaxation with E/A ratio <1.0 and the existence of L-wave. Inferior vena cava was collapsed and estimated right atrial pressure was < 3 mmHg.
After 3.
CASE PRESENTATION
On admission:
Transthoracic echocardiography displayed 50 mm of left atrial diameter, 15 mm of interventricular septal, 14 mm of posterior wall of left ventricle, 40 mm of left ventricular end-diastolic diameter, and 79% of left ventricular ejection fraction. Trans-mitral flow pattern was impaired relaxation with E/A ratio <1.0 and the existence of L-wave. Inferior vena cava was collapsed and estimated right atrial pressure was < 3 mmHg.
Comment 4.
In Section 2.5, was the decreased BNP associated with a decreased creatinine?
Answer 4.
Thank you for your comment. As the condition stabilized, both plasma BNP and serum creatinine decreased. This causal relationship cannot be clearly revealed. We considered that plasma BNP decreased due to the improvement of renal function and/or the decreased LVEDP by improved LVOT obstruction.
Before 4.
CASE PRESENTATION
Following index discharge:
At two month following index discharge, her New York Heart Association function class was II and plasma B-type natriuretic peptide was 578 pg/mL.
After 4.
CASE PRESENTATION
Following index discharge:
At two month following index discharge, her New York Heart Association function class was II, plasma B-type natriuretic peptide was 578 pg/mL, and serum creatinine was 1.06 mg/dL.
Comment 5.
The disappearance of the antibodies over time is curious. Only the result for SS-A was noted. Was SS-B evaluated, as well? SS-B is more specific for Sjogren's syndrome. Are you sure she really had Sjogren's syndrome? Maybe the salivary gland issue was related to something else. Did she have dry eyes, as well?
Answer 5.
We appreciate your comment. SS-B was not elevated. SS-B is more specific for Sjogren syndrome but has low sensitivity. Our patient had symptoms of dry eye without abnormal results of Schirmer test, symptoms of dry mouth, abnormal scintigram of salivary glandular function, and elevated SS-A. According to European-American Consensus Group Modification of the European Community Criteria for Sjogren Syndrome, we consulted rheumatologist and the expert diagnosed her with Sjogren syndrome. We revised our manuscript.
Before 5.
CASE PRESENTATION
Initial diagnosis of PAH:
Although she was diagnosed with Sjogren syndrome due to the existence of anti Ro/SSa antibody and impaired salivary gland function, she had a previous history of steroid-associated necrosis of the talus.
After 5.
CASE PRESENTATION
Initial diagnosis of PAH:
She was diagnosed with Sjogren syndrome due to her symptoms of dry eye and dry mouth, the existence of anti Ro/SSa antibody and impaired salivary gland function, she had a previous history of steroid-associated necrosis of the talus.
Comment 6.
Cibenzoline is not commonly used in the U.S. in the current era for HCM. Disopyramide is more commonly used. Cibenzoline can prolong the QRS and QT interval. Please provide the effect of the medication on these ECG parameters. Is this medication commonly used by the authors for this condition?
Answer 6.
We appreciate your comment. As you pointed out, it is a very important clinical point. Outside of Japan, disopyramide is often used, whereas in Japan cibenzoline is often used, because cibenzoline has less anticholinergic activity than disopyramide and results in continuous reduction in the LV pressure gradient. In our case, QT interval was extended from 479 to 491 millisecond and corrected QT interval was extended from 436 to 482 millisecond after cibenzoline administration. We added the below sentence to the manuscript.
Before 6.
CASE PRESENTATION
Following index discharge:
None.
After 6.
CASE PRESENTATION
Following index discharge:
Although corrected QT interval was extended from 436 to 482 millisecond after cibenzoline administration, further prolongation of corrected QT interval was not observed during follow-up period.
Comment 7.
English editing is recommended.
Answer 7.
Thank you for your comment. A native English speaker reviewed our draft and corrected typographs and grammatical errors.
Before 7.
None.
After 7.
Please see our revised manuscript.

Reviewer 2 Report
The paper by Izumida et al describes a case report of an elderly (74 yrs') women with PAH due to a Sjögren syndrome. The authors found also an outflow tract obstruction an diagnosed HOCM as well.
The paper is wll written and the methods (right heart measurements, LV pressure measurements are well performed as well as the echocardiography.
HOCM ist a heart disease that ususally gets apparent in he 40th and 50the year of life. This patient is liketly to be too old to suffer from a HOCM. The authors claim, that the LV outflow trat obstruction has gone after a 4 weeks treatment with cibenzoline, an anti-arrhythmic drug that alter sodium channels. This sounds unlikely, unleast ist is not explained in the paper how this should work.
It is very likely, that an 74 years old women has a myocardial hypertophy, but it is unlikely, that this is a HOCM. Tee most likely explanation for the loss of the LV obstruction is volume- and preload management. The LV may be better filled and LV obstruction gets less. I doubt, that this is the effect of the antiarrhythmic drug cibenzoline. On Fig 3 A the arterial LV measurement shows a slingshot, that implies, that the estimated gradient is measured too hight due to a technical problem. Echogradients may be overstimated as well.
Author Response
Reviewer 2
General comment.
The paper by Izumida et al describes a case report of an elderly (74 yrs') women with PAH due to a Sjögren syndrome. The authors found also an outflow tract obstruction an diagnosed HOCM as well.
The paper is wll written and the methods (right heart measurements, LV pressure measurements are well performed as well as the echocardiography.
Response.
We express our great appreciation for the reviewer’s critical suggestions and comments. We attempted our best to answer all the comments and revise our draft. We also corrected typographs and grammatical errors. We believe that the revised draft would be suitable for the publication.
Comment 1.
HOCM ist a heart disease that ususally gets apparent in he 40th and 50the year of life. This patient is liketly to be too old to suffer from a HOCM.
Answer 1.
We appreciate your comment. As you pointed out, we initially suspected hypertensive cardiac hypertrophy, because of age and clinical course. However, our case developed diffuse myocardial hypertrophy, despite of well-controlled blood pressure and treatment with angiotensin II receptor blockers, mineralocorticoid receptor antagonists, and anti-PAH agents that should have theoretically inhibited myocardial hypertrophy. Therefore, we performed endo-myocardial biopsy and evaluated the secondary cardiomyopathy.
Before 1.
None.
After 1.
None.
Comment 2.
The authors claim, that the LV outflow trat obstruction has gone after a 4 weeks treatment with cibenzoline, an anti-arrhythmic drug that alter sodium channels. This sounds unlikely, unleast ist is not explained in the paper how this should work. It is very likely, that an 74 years old women has a myocardial hypertophy, but it is unlikely, that this is a HOCM. Tee most likely explanation for the loss of the LV obstruction is volume- and preload management. The LV may be better filled and LV obstruction gets less. I doubt, that this is the effect of the antiarrhythmic drug cibenzoline.
Answer 2.
Thank you for your comment. Several guidelines from Japan and America described that disopyramide and cibenzoline are effective for reducing the pressure gradient at both rest and during exercise, not through change of preload and afterload, but through the decreased contraction force.
Before 2.
DISCUSSION
How to manage patients with HOCM and PAH:
None.
After 2.
DISCUSSION
How to manage patients with HOCM and PAH:
Drastic change in the preload might manifest occult HOCM. Instead, appropriate adjustment of preload using right heart catheterization and aggressive therapeutic intervention to HOCM had better be prioritized using beta-blocker and cibenzoline, which are effective for reducing the pressure gradient at both rest and during exercise, not through change of preload and afterload, but through the decreased contraction force, as we did.
Comment 3.
On Fig 3 A the arterial LV measurement shows a slingshot, that implies, that the estimated gradient is measured too hight due to a technical problem. Echogradients may be overstimated as well.
Answer 3.
Thank you for your comment. We partially agree with you. As you pointed out, the measured gradient of LVOT might be overestimated. However, we performed several measurements of LV pressure and found similar waveforms. Although there might be a problem with the examination, we wondered if the left ventricular morphology was affected. Similarly, with regard to echocardiography, expert sonographers performed the examination, and other blood flow separations are being performed well, so we think that technical problems are unlikely.
Before 3.
None.
After 3.
CASE PRESENTATION
Following the initiation of anti-PAH agent:
One month after the termination of anti-PAH agents, a transthoracic echocardiography found an accelerated blood flow in the left ventricular outflow tract with Vmax 3.4 m/sec with flow separation and moderate mitral regurgitation due to systolic anterior motion of mitral valve (Figure 2A).

Reviewer 3 Report
In this study entitled “Therapeutic strategy for a patient with concomitant pulmonary artery hypertension and hypertrophic obstructive cardiomyopathy; a rare case report”, the authors reported a rare case of Sjogren syndrome-associated pulmonary artery hypertension (PAH) with hypertrophic obstructive cardiomyopathy (HOCM). The concomitant occurrence of these two diseases is very rare and, previously, few studies had reported similar patients with concomitant PAH and HOCM. Therefore, this study can provide insights for physicians regarding the possible management of patients with similar presentations. Overall, the manuscript is well-written, has informative illustrations, and proposes a management strategy for patients with concomitant PAH and HOCM. Some minor points regarding this study are:
1. It is suggested to discuss the option of septal reduction therapies (i.e., surgical myomectomy or alcohol septal ablation) for HOCM-related LVOT obstruction in patients with concomitant PAH and HOCM.
2. In the case presentation, for the ease of readers, it is recommended to describe the level of dyspnea at each time interval based on the NYHA classification.
Author Response
Reviewer 3
General comment.
In this study entitled “Therapeutic strategy for a patient with concomitant pulmonary artery hypertension and hypertrophic obstructive cardiomyopathy; a rare case report”, the authors reported a rare case of Sjogren syndrome-associated pulmonary artery hypertension (PAH) with hypertrophic obstructive cardiomyopathy (HOCM). The concomitant occurrence of these two diseases is very rare and, previously, few studies had reported similar patients with concomitant PAH and HOCM. Therefore, this study can provide insights for physicians regarding the possible management of patients with similar presentations. Overall, the manuscript is well-written, has informative illustrations, and proposes a management strategy for patients with concomitant PAH and HOCM. Some minor points regarding this study are:
Response.
We express our great appreciation for the reviewer’s critical suggestions and comments. We attempted our best to answer all the comments and revise our draft. We also corrected typographs and grammatical errors. We believe that the revised draft would be suitable for the publication.
Comment 1.
It is suggested to discuss the option of septal reduction therapies (i.e., surgical myomectomy or alcohol septal ablation) for HOCM-related LVOT obstruction in patients with concomitant PAH and HOCM.
Answer 1.
We appreciate your comment. We agree with you. Of course, surgical septal myectomy and alcohol septal ablation should be considered for symptomatic drug-resistant HOCM. As described in our manuscript, we considered the improvement of HOCM is prioritized for patients with concomitant PAH and HOCM. Therefore, septal reduction therapy should be aggressively considered in cases with patients with HOCM and LVOTO refractory to medical therapy including beta blockers and Na+ channel blockers. Generally, in younger patients, surgical intervention would be preferable, but alcohol septal ablation should be considered in patients with severe PAH.
Before 1.
DISCUSSION
How to manage patients with HOCM and PAH:
None.
After 1.
DISCUSSION
How to manage patients with HOCM and PAH:
For symptomatic drug-resistant HOCM patients, septal reduction therapy should be aggressively considered. Generally, in younger patients, surgical intervention would be preferable. Alcohol septal ablation is alternative in older patients or those with severe PAH.
Comment 2.
In the case presentation, for the ease of readers, it is recommended to describe the level of dyspnea at each time interval based on the NYHA classification.
Answer 2.
Thank you for your comment. We added the data to our manuscript.
Before 2.
None.
After 2.
Please see our revised manuscript.
